# Relationship between Personality and Mortality among Japanese Older Adults: A 14-Year Longitudinal Study

**DOI:** 10.3390/ijerph19042413

**Published:** 2022-02-19

**Authors:** Hajime Iwasa, Hiroki Inagaki, Yukie Masui, Yasuyuki Gondo

**Affiliations:** 1Department of Public Health, School of Medicine, Fukushima Medical University, Fukushima 960-1295, Japan; 2Tokyo Metropolitan Institute of Gerontology, Tokyo 173-0015, Japan; inagaki@tmig.or.jp (H.I.); ymasui@tmig.or.jp (Y.M.); y.gondo.hus@osaka-u.ac.jp (Y.G.); 3Graduate School of Human Sciences, Osaka University, Osaka 565-0871, Japan

**Keywords:** all-cause mortality, Big Five personality, community-dwelling older adults, NEO Five-Factor Inventory

## Abstract

Personality is one of the fundamental factors in determining longevity. We used a 14-year mortality surveillance to investigate the relationship between the Big Five personality traits and all-cause mortality among older adults dwelling in a Japanese community. Individuals over 65 years old (484 males and 743 females) were recruited for the study. We used the NEO Five-Factor Inventory to assess the Big Five personality traits: neuroticism, extraversion, openness, agreeableness, and conscientiousness. During the follow-up period, 502 persons (250 men and 252 women) had died. Cox proportional hazards regression controlling for covariates showed that extraversion (hazard ratio [HR] = 0.783, 95% confidence interval [CI] = 0.636 to 0.965 and HR = 0.757, 95% CI = 0.607 to 0.944 for the middle and highest tertiles, respectively), openness (HR = 0.768, 95% CI = 0.608 to 0.969 for the highest tertile), and conscientiousness (HR = 0.745, 95% CI = 0.607 to 0.913 and HR = 0.667, 95% CI = 0.530 to 0.840 for the middle and highest tertiles, respectively) were inversely associated with mortality when the five traits were analyzed separately. Our findings suggest that older adults who have a higher level of either extraversion, openness, or conscientiousness are more likely to live longer.

## 1. Introduction

According to a previous proposal [1], personality is the fourth most important determinant for longevity, after genetic, physical, and biological elements. “Personality refers to the psychological qualities that contribute to an individual’s enduring and distinctive patterns of feeling, thinking, and behaving” [2]. The Big Five personality consists of five major domains of personality: neuroticism, extraversion, openness, agreeableness, and conscientiousness [3]. It is currently the most widely used framework, even in the health sciences. The association of personality with longevity has been explored by many studies [4,5,6,7,8,9,10,11,12,13,14,15,16].

Neuroticism refers to the tendency of being vulnerable to psychological distress. Previous studies have presented a variety of findings on the association of neuroticism with mortality; this also includes the findings indicating no relationship between the two [10,11,12], that neuroticism is associated with an accelerated risk of death [5,6,16,17], and that neuroticism is related to a decreased risk of mortality [7,8]. People with a higher tendency of neuroticism tend to have more negative emotional experiences, which have been associated with an accelerated risk of death [4,18,19]. Conversely, neuroticism may have a role in mitigating mortality risk, as individuals are prone to consult a doctor owing to their greater neuroticism [20]. Therefore, given these mixed effects, the neuroticism–longevity relationship remains unclear.

Extraversion refers to the disposition to be sociable and have positive emotional experiences. Individuals who are highly extraverted tend to be involved in social interactions, which reduces the effects of stress [21], thereby buffering individuals against many of the challenges of old age (e.g., disability and bereavement) [22]. Thus, extraversion can be associated with a decreased risk against mortality through its involvement in social interactions.

Openness refers to intellectual curiosity and a preference for varied experiences. A previous study reported a significant association of openness with mortality among older adults [13]. Additionally, Masui et al. [14] revealed that a high level of openness was observed in Japanese centenarians (i.e., a person who has reached the age of 100 years). It has been speculated that higher intellectual curiosity, which is close to openness, helps older adults adapt to the many health challenges in old age. Thus, openness can be associated with a decreased risk of death.

Agreeableness is defined as the disposition of being trusting, compassionate, and cooperative. Previous studies have shown varying findings on the association of agreeableness with mortality (including studies showing no relationship [6,13,15] and those showing that agreeableness is protective against mortality [7]). Highly agreeable people are socially desirable and psychologically healthy because they tend to empathize with others [23,24] and have effective social support that aids in reducing the impact of stress [21,25]. Therefore, people with high agreeableness may experience health and longevity benefits. However, since the relationship between agreeableness and health issues among older adults has not been well reported thus far, the association of agreeableness with mortality is unclear.

Conscientiousness makes a person diligent, organized, and achievement-oriented, and is considered as the most important personality trait in relation to longevity [6,7,9]. Highly conscientious individuals tend to have the advantage of being more committed to health-related behaviors such as cigarette avoidance [26], drinking moderately [27], regular physical fitness [28], and faithful compliance with medical advice [9,17], all of which can reduce the risk of mortality. Thus, conscientiousness can be related to a decreased risk of mortality through such healthy behaviors.

We have previously reported on the relationship between the Big Five personality factors and longevity among Japanese older adults using a 5-year mortality surveillance [29]. Our previous study found an inverse association of three traits: extraversion, openness, and conscientiousness, with mortality. In the previous examination using the 5-year follow-up [29], the number of events that occurred during the follow-up (deaths) was somewhat low (127/1228 = 10.3%). Therefore, it is possible that some of the associations between personality and mortality were unclear (e.g., the association of openness and mortality) [29]. Additionally, it will take a considerable amount of time for the association between personality and mortality to become clear because possible mediating mechanisms for the association, which include incidences of lifestyle-related diseases and functional decline, may be considered as gradually progressing throughout old age and eventually affecting mortality over a long period of time. Therefore, it was necessary to extend the follow-up period and re-examine this association throughout old age to discover more reliable findings in examining the relationship.

Thus, in the present study, we used a 14-year mortality surveillance to investigate the relationship between the Big Five personality traits and all-cause mortality among older adults dwelling in a Japanese community. We postulated that among the five domains of personality, extraversion, openness, and conscientiousness would be associated with a decreased risk for death, and the results would be a replication of the previous findings [29], even if the follow-up period was extended.

## 2. Materials and Methods

### 2.1. Participants

The present study used the Longitudinal Interdisciplinary Study on Aging conducted by the Tokyo Metropolitan Institute of Gerontology as the source of data [29,30]. A sample of 4440 residents (aged 50–74 years) were obtained systematically from the resident registration files of City A, located in the northern part of Tokyo, from 1991. The 3097 sets of data were acquired during the first visit in 1991, and follow-up interviews were conducted annually thereafter until 2000 [30]. Of the 1749 individuals who participated in the 2000 survey, only the data set of 1233 individuals over 65 years old was used. Of these subjects, two individuals with missing values on the NEO Five-Factor Inventory (NEO-FFI) and four individuals with unspecified years of education were excluded from the analysis. After these exclusions, the remaining 1227 individuals (484 men and 743 women) were included in the prospective cohort study. Their mortality was then checked during a 14-year surveillance. The study was approved by the Ethics Committee of the Tokyo Metropolitan Institute of Gerontology (approval number: R21-064). Before conducting the home-visit survey, we provided the participants with the study protocol and explained the following instructions: First, the participants would be completely free to decide whether to participate in the study; second, they could withdraw from the study at any time; third, the participant would not experience any negative effects if they chose to not participate in the study or withdrew from it at any time; fourth, the participants’ personal information would be protected. Responding to the questionnaires during the home-visit survey after the participants were provided with the information above, was considered as confirmation that they agreed with the purpose of the survey and were voluntarily participating. We were provided access to the municipal resident registration files by the City A authorities.

### 2.2. Mortality Surveillance

The baseline for surveillance was kept as 1 January 2001 since the last survey was completed at the end of 2000. Thereafter, a 14-year mortality surveillance was conducted from 1 January 2001 to 1 July 2015. The current residence of the participants in City A, as of 1 July 2015, was ascertained using the municipality’s resident registration files. Residents’ dates of relocation or death were determined from the registry files and were used to compute survival time. The certification of all decedents and of those who moved were obtained from the authorities of City A. The proof for all the decedents and the relocated persons was received from the government of City A. The outcome variable in the analysis was survival time, which was computed as the number of days from the baseline to the date of death or censoring (including survivors and those who dropped out due to migration from City A).

### 2.3. Personality Traits

We administered the Japanese version of the NEO-FFI [23,24] to measure the five major domains of personality traits in 2000. The reliability and validity of the scale have been confirmed [24]. Participants were asked to rate each of the 60 items on a five-point Likert scale, from “strongly disagree” to “strongly agree”. A total score (range: 0–48) for each trait was obtained by summing the scores for each item (range: 0–4). The higher the score, the higher the level of traits. Cronbach’s alpha coefficients, which indicated the reliability of each domain of personality, for neuroticism, extraversion, openness, agreeableness, and conscientiousness in the current study, were 0.785, 0.770, 0.578, 0.740, and 0.821, respectively. Three categories (lowest, medium, and highest tertiles) were created, corresponding to approximating the three tertiles of the total score for each of the five traits.

### 2.4. Covariates

The baseline data on age, gender, the number of years of education, living arrangement (dichotomized: living alone or living with family), psychiatric problems, chronic diseases, and dependence in instrumental activities of daily living (IADL) were used as covariates to examine the independent association of personality with mortality. Psychiatric problems and chronic diseases were self-rated by the participants. Chronic diseases referred to having at least one of the following diseases: cancer, diabetes, heart disease, and stroke. Participants were asked to assess independence in terms of five daily IADL tasks (e.g., use of public transport and food preparation) [31]. The higher the score, the higher the functional level of IADL. In this study, a 4/5 cutoff score (i.e., scores less than 4 were classified as IADL dependent) was used to determine whether a participant was dependent with respect to IADL [32].

### 2.5. Statistical Analysis

To determine the differences in the characteristics between the two groups of survivors/dropouts and non-survivors, a *t*-test was conducted for continuous variables and a chi-squared test for categorical variables.

Cox proportional hazards models, adjusted for the aforementioned covariates, were used to examine the independent relationships between each personality trait and mortality. In this analysis, we conducted two regression models: model 1 (entering the NEO-FFI five factors into the model separately) and model 2 (entering the NEO-FFI five factors into the model simultaneously). All probability values were two-tailed. The significance level in all statistical tests was set at 5%. IBM SPSS version 27 (IBM Corporation, Armonk, NY, USA) was used for the analysis.

## 3. Results

After 14.5 years, 502 (40.9%, 250 men and 252 women) of the 1227 adults had died and 177 people (14.4%, 62 men and 115 women) had migrated from the target area. Table 1 demonstrates the characteristics of the participants and the distributed scores of each personality scale collected at baseline. The differences in those characteristics between the two groups were compared. Non-survivors (i.e., those who died) were more likely to be older (*p* < 0.001), were men (*p* < 0.001), had fewer years of education, had chronic diseases (*p* < 0.001), and had an IADL dependence (*p* < 0.001) compared to survivors/dropouts. Moreover, non-survivors were more likely to score lower on extraversion (*p* < 0.001), openness (*p* < 0.001), agreeableness (*p* = 0.037), and conscientiousness (*p* < 0.001) compared to survivors/dropouts.

Table 2 shows the associations between personality and mortality. Cox proportional hazards models showed that extraversion (hazard ratio [HR] = 0.781, 95% confidence interval [CI] = 0.634 to 0.962 and HR = 0.756, 95% CI = 0.606 to 0.944 for the middle and highest tertiles, respectively), openness (HR = 0.768, 95% CI = 0.608 to 0.969 for the highest tertile), and conscientiousness (HR = 0.747, 95% CI = 0.609 to 0.916 and HR = 0.669, 95% CI = 0.532 to 0.842 for the middle and highest tertiles, respectively) were inversely associated with all-cause mortality when analyzing the five factors separately. Neither neuroticism nor agreeableness were related to mortality. Only conscientiousness was independently and inversely associated with mortality when the traits were analyzed simultaneously (HR = 0.762, 95% CI = 0.615 to 0.944 and HR = 0.715, 95% CI = 0.545 to 0.938 for the middle and highest tertiles, respectively).

The results considering the scores of personality traits as continuous variables are discussed below. Out of the five traits, extraversion (HR = 0.971, 95% CI = 0.957 to 0.986 for a one-point increase), openness (HR = 0.977, 95% CI = 0.959 to 0.996), and conscientiousness (HR = 0.970, 95% CI = 0.957 to 0.984) were inversely associated with mortality when analyzed separately. Only conscientiousness was independently and inversely associated with mortality when analyzed simultaneously (HR = 0.978, 95% CI = 0.960 to 0.996 for a one-point increase).

## 4. Discussion

The present study tested the association of the Big Five personality traits with all-cause mortality in older adults dwelling in a community in Japan, using a 14-year mortality surveillance. Among the traits, extraversion, openness, and conscientiousness were associated with lower mortality, echoing the findings of our previous study [29].

Our results demonstrate that conscientiousness acts as a protective factor for mortality, which is in accordance with previous investigations [6,7,9,13,15,29]. High conscientious is expected to reduce the risk of mortality for particular reasons. First, self-discipline, a subscale of conscientiousness, has been reported to be protectively associated with mortality. [7]. In addition, conscientious persons were characterized as being more self-controlled [33]. People with such tendencies may have the advantage of being more willing to engage in a variety of favorable health-related behaviors [34]. Second, highly conscientious people tend to plan for the long term, easing their process of preparation to avoid daily frustrations (e.g., by preparing in advance for an important job or preparing for future risks, such as purchasing life insurance) [9]. Third, older individuals who have higher levels of conscientiousness are less likely to develop disability [35], which is strongly associated with mortality [36].

Furthermore, the present study also pointed out that only conscientiousness was independently associated with mortality when the five traits were analyzed simultaneously (i.e., model 2). This finding was consistent with that of a previous meta-analytic study [15]. In addition, another meta-analytic study reported that conscientiousness had the strongest association to mortality compared to other personality factors, intelligence, and socio-economic status [37]. Thus, previous findings and our findings suggest that conscientiousness is one of the most important personality traits, which acts protectively for longevity.

Extraversion was identified as protective against mortality in the present study, concordant with the findings of previous research [6,16]. There are several reasons why extraversion may reduce the risk of mortality. First, higher extraversion implies more optimism and is associated with an external attributional style [33], which may mitigate the harmful consequences of stress. Second, individuals with higher extraversion tend to be more socially involved, which may diminish the adverse effects of stress [21] and guard against a number of hardships in old age [22].

Openness was inversely associated with mortality in the present study. The finding was similar to those of previous studies [13,14]. The previous findings and our findings suggest that openness also may be a longevity-related personality trait. Openness may reduce mortality risk for various reasons. First, people with high openness tend to maintain cognitive function [38], which is related to participation in health surveys [39] and health examinations [40] for older adults. This may be because they take interest in their health as they are sensitive to information about health promotion. Second, people with high openness are prone to like a variety of new experiences, such as actively trying new activities and new foods [2,3]. Thus, individuals with a higher level of openness are more likely to take positive new actions based on health information transmitted through various media, people, and government agencies, among others. Furthermore, this tendency is closely related to health literacy [41]. Openness is also closely correlated to health literacy [42].

The current research has some limitations. First, psychiatric problems and chronic diseases were self-rated by the participants. Since self-rated health assessments may not be as accurate as evaluations conducted by medical professionals, the measurement may be unreliable. Second, key confounders such as tobacco use, drinking alcohol, and exercise could not be employed as covariates in analyzing the independent association between personality and mortality as no information about these variables was collected at baseline.

## 5. Conclusions

Our study—conducted during a 14-year follow-up period among Japanese older adults—found that of the Big Five personalities, extraversion, openness, and conscientiousness were inversely associated with all-cause mortality. Additionally, it was found that conscientiousness is one of the most important personality traits, which acts protectively for longevity. The present study’s findings may be useful in developing health strategies for longevity among older adults. Personality traits that are detrimental to longevity are not easily modified drastically through interventions such as psychotherapy. Therefore, people with such adverse personality traits need to be advised to enhance their performance on other potentially modifiable components of longevity such as adverse health behaviors and lifestyle-related diseases to offset the negative effects of personality on longevity. For instance, persons with a low level of conscientiousness may be recommended to quit unhealthy behaviors, such as tobacco use and excessive drinking, and increase frequent exercise. In individuals with a low level of extraversion, the usage of social welfare services may be promoted to counteract the shortage of social interaction. Finally, for those with a low level of openness, it may be useful to facilitate their acquisition of health literacy skills.

## Figures and Tables

**Table 1 ijerph-19-02413-t001:** Distribution of participants’ characteristics and personality scores at baseline (non-survivors vs. survivors) (*n* = 1227).

	Deceased (*n* = 502)	Survivors/Dropouts (*n* = 725)	*p* *
Age (mean years ± SD)	74.6 ± 5.0	71.0 ± 4.6	<0.001
Gender (% women)	252 (50.2)	491 (67.7)	<0.001
Number of years of education (years ± SD)	10.2 ± 3.0	10.7 ± 2.6	<0.001
Living arrangement (alone), n (%)	78 (15.5)	114 (15.7)	0.930
Psychiatric problems, n (%)	11 (2.2)	10 (1.4)	0.281
Chronic diseases, n (%)	148 (29.5)	108 (14.9)	<0.001
Instrumental activities of daily living dependence, n (%)	138 (27.5)	96 (13.2)	<0.001
Neuroticism (mean scores ± SD)	16.7 ± 6.9	17.0 ± 6.3	0.545
Extraversion (mean scores ± SD)	25.5 ± 6.3	27.3 ± 5.7	<0.001
Openness (mean scores ± SD)	23.4 ± 5.0	24.9 ± 4.9	<0.001
Agreeableness (mean scores ± SD)	33.8 ± 5.1	34.3 ± 5.0	0.037
Conscientiousness (mean scores ± SD)	30.3 ± 6.7	31.9 ± 5.9	<0.001

* *t*-tests for continuous variables and χ² tests for categorical variables were used to elucidate the differences in the properties between the two groups (non-survivors vs. survivors). SD = standard deviation.

**Table 2 ijerph-19-02413-t002:** Associations of personality traits with all-cause mortality (N = 1227).

				Model 1 ^a,c^		Model 2 ^b,c^	
	Tertile	N	Deaths (%)	Hazard Ratio (95% Confidence Interval)	*p*	Hazard Ratio (95% Confidence Interval)	*p*
Neuroticism	Lowest (ref.)	456	183 (40.1)	1		1	
	Middle	360	155 (43.1)	1.156 (0.932–1.433)	0.188	1.108 (0.875–1.402)	0.396
	Highest	411	164 (39.9)	1.206 (0.970–1.500)	0.092	1.094 (0.850–1.406)	0.470
Extraversion	Lowest (ref.)	436	211 (48.4)	1		1	
	Middle	416	157 (37.7)	0.781 (0.634–0.962)	0.020	0.825 (0.663–1.025)	0.083
	Highest	375	134 (35.7)	0.756 (0.606–0.944)	0.013	0.897 (0.690–1.165)	0.414
Openness	Lowest (ref.)	411	203 (49.4)	1		1	
	Middle	415	163 (39.3)	0.828 (0.670–1.022)	0.078	0.833 (0.674–1.030)	0.092
	Highest	401	136 (33.9)	0.768 (0.608–0.969)	0.027	0.813 (0.640–1.032)	0.089
Agreeableness	Lowest (ref.)	458	191 (41.7)	1		1	
	Middle	409	171 (41.8)	1.040 (0.844–1.281)	0.712	1.183 (0.949–1.475)	0.135
	Highest	360	140 (38.9)	0.898 (0.718–1.122)	0.342	1.124 (0.863–1.466)	0.386
Conscientiousness	Lowest (ref.)	463	222 (47.9)	1		1	
	Middle	423	161 (38.1)	0.747 (0.609–0.916)	0.005	0.762 (0.615–0.944)	0.013
	Highest	341	119 (34.9)	0.669 (0.532–0.842)	<0.001	0.715 (0.545–0.938)	0.015

^a^ Personality traits were analyzed individually using regression models; ^b^ personality traits were analyzed simultaneously with the help of regression models; ^c^ controlling for age, gender, number of years of education, living arrangement, psychiatric problems, and chronic diseases, and dependence in instrumental activities of daily living.

## Data Availability

These data are not appropriate for public disclosure due to ethical issues. If you are a researcher who is interested in analysis using these data, please request access to confidential data from the Ethics Committee of the Tokyo Metropolitan Institute of Gerontology.

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
