# Peer review of "Relationship between Personality and Mortality among Japanese Older Adults: A 14-Year Longitudinal Study"

_ijerph, 2022, doi:10.3390/ijerph19042413_

Round 1
Reviewer 1 Report
The presented manuscript is a continuation of the authors' work on the study of the association of Five personality traits with the risk of death in the elderly. The article summarizes observational data for a 14-year period. The results have a high social significance and are of interest to a wide range of readers. The manuscript is written in an understandable language, statistical analysis seems to be sufficient for the conclusions drawn. The reliability of the results obtained is confirmed by the similarity of the results in the authors' article, which were presented as a result of monitoring the same population for 5 years. However, there are a number of remarks that must be taken into account before the publication of the manuscript.
Line 180: The numbers do not correspond to those given in Table 2.
Lines 183-186: It is not clear whether the text and the presented figures refer to this article or to some other experiment. There are no such numerical data in the tables of the article under review. Please check both the text and digital information.
There are several confusing phrases in the discussion. These phrases are written as if there is a direct, not an inverse, association with mortality:
Lines 200-201: self-discipline, which is a subscale 200 of conscientiousness, is reported to be a predictor of mortality
Lines 207-208: a higher level of conscientiousness is related to a reduced 207 risk of incident disability [34], which is strongly associated with mortality
Lines 212-213: conscientiousness had the strongest association to mortality compared to other personality factors
Lines 214-215: conscientiousness is the most reliable predictor of mortality
Lines 246-247: was found that conscientiousness was the most reliable predictor of mortality.
Author Response
Reviewer: 1
Thank you for reviewing our manuscript. We have revised the manuscript in accordance with the comments from the reviewer. The following are our responses to the reviewer’s comments. The revised parts are underlined in red.
Comment #1
The presented manuscript is a continuation of the authors' work on the study of the association of Five personality traits with the risk of death in the elderly. The article summarizes observational data for a 14-year period. The results have a high social significance and are of interest to a wide range of readers. The manuscript is written in an understandable language, statistical analysis seems to be sufficient for the conclusions drawn. The reliability of the results obtained is confirmed by the similarity of the results in the authors' article, which were presented as a result of monitoring the same population for 5 years. However, there are a number of remarks that must be taken into account before the publication of the manuscript.
(Response #1)
Thank you for reviewing our manuscript. I made a revision in accordance with your comments.
Comment #2
Line 180: The numbers do not correspond to those given in Table 2.
(Response #2)
Thank you for pointing this out. I correct the mistake in the Table 2.
Comment #3
Lines 183-186: It is not clear whether the text and the presented figures refer to this article or to some other experiment. There are no such numerical data in the tables of the article under review. Please check both the text and digital information.
(Response #3)
Thank you for pointing this out. I correct the mistake in the text body as follows (Line208):
[Only conscientiousness was independently and inversely associated with mortality when analyzed simultaneously (HR = 0.978, 95% CI = 0.960 to 0.996 for a one point increase).]
Comment #4
There are several confusing phrases in the discussion. These phrases are written as if there is a direct, not an inverse, association with mortality:
Lines 200-201: self-discipline, which is a subscale 200 of conscientiousness, is reported to be a predictor of mortality
(Response #4)
As per the comment, we modified the phrase as follows (Line223):
[First, self-discipline, a subscale of conscientiousness, has been reported to be protectively associated with mortality.]
Comment #5
Lines 207-208: a higher level of conscientiousness is related to a reduced 207 risk of incident disability [34], which is strongly associated with mortality
(Response #5)
As per the comment, this sentence is taken from previous studies. People with higher levels of conscientiousness were less likely to experience functional disability [Krueger et al., 2006]. In addition, previous studies showed that the functional disability was strongly associated with mortality [Ramos et al., 2001].
Comment #6
Lines 212-213: conscientiousness had the strongest association to mortality compared to other personality factors
(Response #6)
Regarding the comment. This phrase from a previous meta-analytic study that found conscientiousness had the strongest association with mortality compared to other personality factors, intelligence, and socio-economic status. Thus I would like to use this part as it is, without changing it.
Comment #7
Lines 214-215: conscientiousness is the most reliable predictor of mortality
(Response #7)
As per the comment, we modified the phrase as follows (Line238):
[Thus, previous findings and our findings suggest that conscientiousness is one of the most important personality traits, which acts protectively for longevity.]
Comment #8
Lines 246-247: was found that conscientiousness was the most reliable predictor of mortality.
(Response #8)
As per the comment, we modified the phrase as follows (Line272):
[Additionally, it was found that conscientiousness is one of the most important personality traits, which acts protectively for longevity.]

Reviewer 2 Report
The manuscript titled “Relationship between personality and mortality among Japanese older adults: A 14-year longitudinal study” proposed an investigation on the relationship between personality factors and all-cause mortality in a large sample (484 men and 743 women) of community-dwelling Japanese older adults. The NEO Five Factor inventory was used to assess the big five personality traits. Over the study period, 502 persons died. Cox hazard models were used to analyze data. Results showed that extraversion, openness and conscientiousness were associated in an inverse manner with mortality, when the traits were analyzed separately. Authors discussed their results in light of previous literature.
I carefully read the manuscript, and I think it may be of interest for the readers of International Journal of Environmental Research and Public Health. The manuscript is well-written and properly addresses the interesting issue of which personality factor(s) seem(s) to be associated with mortality ratio. This topic has a long tradition of studies, some of them with contradicting results, and more knowledge coming from well-conducted primary studies is needed. I found that the introduction section and the methodology employed are clear and detailed, as well as the explanations provided in the discussion section. I only have few minor remarks:
Abstract section
line 24: I would not use the word “predictors” since it is difficult to support the idea that a single personality factor could predict longevity. Indeed, statistically it’s a predictor, but theoretically speaking it is a variable could be considered as associated to a greater likelihood of living longer.
Then, I would use the expression “are associated to”.
Introduction section
lines 79-80: you wrote that “Therefore, it was necessary to extend the follow-up period and re-examine this association throughout old age.” It would be useful to explain why it is necessary to extend the follow-up period, since I think that the answer to this question constitutes the rationale of the study. In other words, what is the added value for a longer follow-up period with respect to the previous study?
line84-85: again, you wrote that “the results will be a reproduction of the previous findings”. So, where is the novelty? Why this research should be interesting if we know already the results from a previous study?
Materials and Methods section
lines 117-118: please explain when the Japanese of the NEO-FFI was administered.
lines 125-126: please explain the choice to divide the total score for each trait into three categories.
Results section
lines 154-155: the percentage of 10.3% does not correspond to 502 / 1227 * 100 = 40,9%
line 186: the last sentence is incomplete.
Discussion section
line 215: again, I would not say that “conscientiousness is the most reliable predictor of mortality among older adults”, but rather than there is an association between conscientiousness and mortality.
Conclusions section
The whole section seems to be another small abstract. I suggest to broad the view of your results arguing what is the importance of your study, how the results could be useful e.g. to improve people life or to reduce national healthcare system expenditure.
Author Response
Reviewer: 2
Thank you for reviewing our manuscript. We have revised the manuscript in accordance with the comments from the reviewer. The following are our responses to the reviewer’s comments. The revised parts are underlined in red.
Comment #1
The manuscript titled “Relationship between personality and mortality among Japanese older adults: A 14-year longitudinal study” proposed an investigation on the relationship between personality factors and all-cause mortality in a large sample (484 men and 743 women) of community-dwelling Japanese older adults. The NEO Five Factor inventory was used to assess the big five personality traits. Over the study period, 502 persons died. Cox hazard models were used to analyze data. Results showed that extraversion, openness and conscientiousness were associated in an inverse manner with mortality, when the traits were analyzed separately. Authors discussed their results in light of previous literature.
I carefully read the manuscript, and I think it may be of interest for the readers of International Journal of Environmental Research and Public Health. The manuscript is well-written and properly addresses the interesting issue of which personality factor(s) seem(s) to be associated with mortality ratio. This topic has a long tradition of studies, some of them with contradicting results, and more knowledge coming from well-conducted primary studies is needed. I found that the introduction section and the methodology employed are clear and detailed, as well as the explanations provided in the discussion section. I only have few minor remarks:
(Response #1)
Thank you for reviewing our manuscript. I made a revision in accordance with your comments.
Comment #2
Abstract section
line 24: I would not use the word “predictors” since it is difficult to support the idea that a single personality factor could predict longevity. Indeed, statistically it’s a predictor, but theoretically speaking it is a variable could be considered as associated to a greater likelihood of living longer.
Then, I would use the expression “are associated to”.
(Response #2)
As per the comments, we modified the last sentence of the abstract as follows:
[Our findings suggest that older adults who have a higher level of either extraversion, openness, or conscientiousness are more likely to live longer.]
Comment #3
Introduction section
lines 79-80: you wrote that “Therefore, it was necessary to extend the follow-up period and re-examine this association throughout old age.” It would be useful to explain why it is necessary to extend the follow-up period, since I think that the answer to this question constitutes the rationale of the study. In other words, what is the added value for a longer follow-up period with respect to the previous study?
line84-85: again, you wrote that “the results will be a reproduction of the previous findings”. So, where is the novelty? Why this research should be interesting if we know already the results from a previous study?
(Response #3)
As per the comment, we added the information regarding the reasons for using the longer follow-up period in the study as follows (Line 81):
[In the previous examination using the 5-year follow-up [Iwasa et al., 2008], the number of events that occurred during the follow-up (deaths) was somewhat low (127/1228 = 10.3%). Therefore, it is possible that some of the associations between personality and mortality were unclear (e.g. the association of openness and mortality) [Iwasa et al., 2008]. Additionally, it will take a considerable amount of time for the association between personality and mortality to become clear because possible mediating mechanisms for the association, which include incidences of lifestyle-related diseases and functional decline, may be considered as gradually progressing throughout old age and eventually affecting mortality over a long period of time. Therefore, it was necessary to extend the follow-up period and re-examine this association throughout old age to discover more reliable findings in examining the relationship.]
In addition, as per the comment, we modified the last sentence of the introduction section as follows (Line 94):
[We postulated that among the five domains of personality, extraversion, openness, and conscientiousness would be associated with a decreased risk for death, and the results would be a replication of the previous findings [Iwasa et al., 2008], even if the follow-up period is extended.]
Comment #4
Materials and Methods section
lines 117-118: please explain when the Japanese of the NEO-FFI was administered.
(Response #4)
As per the reviewer’s comment, I added information regarding the NEO-FFI was administered in the text body as follows (Line 139):
[We administered the Japanese version of the NEO-FFI [23,24] to measure the five major domains of personality traits in 2000.]
Comment #5
lines 125-126: please explain the choice to divide the total score for each trait into three categories.
(Response #5)
With respect to the comment, we used the tertiles of the total score for each of the five personality traits, based on our previous study [Iwasa et al., 2008]. The reason for using the tertiles for the analysis is that the readers can more easily understand the linear relationship between personality and mortality. It would have been preferable to have more categories, such as quintiles, but since the sample size in the study was not relatively large, the study used the tertiles. In epidemiological studies, continuous variables are often divided into tertiles or quartiles for analysis. The results using personality scores as continuous variables are also given in the text body.
Comment #6
Results section
lines 154-155: the percentage of 10.3% does not correspond to 502 / 1227 * 100 = 40,9%
(Response #6)
Thank you for pointing this out. I correct the mistake.
Comment #7
line 186: the last sentence is incomplete.
(Response #7)
I've filled in the missing information as follows (Line 207):
[Only conscientiousness was independently and inversely associated with mortality when analyzed simultaneously (HR = 0.978, 95% CI = 0.960 to 0.996 for a one point increase).]
Comment #8
Discussion section
line 215: again, I would not say that “conscientiousness is the most reliable predictor of mortality among older adults”, but rather than there is an association between conscientiousness and mortality.
(Response #8)
As per the comment, we modified the phrase as follows (Line 238):
[Thus, previous findings and our findings suggest that conscientiousness is one of the most important personality traits, which acts protectively for longevity.]
Comment #9
Conclusions section
The whole section seems to be another small abstract. I suggest to broad the view of your results arguing what is the importance of your study, how the results could be useful e.g. to improve people life or to reduce national healthcare system expenditure.
(Response #9)
Thanks a lot for your suggestion. We modified the conclusion section as follows (Line 270):
[Our study—conducted during a 14-year follow-up period among Japanese older adults—found that of the Big Five personalities, extraversion, openness, and conscientiousness were inversely associated with all-cause mortality. Additionally, it was found that conscientiousness is one of the most important personality traits, which acts protectively for longevity. The present study’s findings may be useful in developing health strategies for longevity among older adults. Personality traits that are detrimental to longevity are not easily modified drastically through interventions such as psychotherapy. Therefore, people with such adverse personality traits need to be advised to enhance their performance on other potentially modifiable components of longevity such as adverse health behaviors and lifestyle-related diseases to offset the negative effects of personality on longevity. For instance, persons with a low level of conscientiousness may be recommended to quit unhealthy behaviors, such as tobacco use and excessive drinking, and increase frequent exercise. In individuals with a low level of extraversion, the usage of social-welfare services may be promoted to counteract the shortage of social interaction. Finally, for those with a low level of openness, it may be useful to facilitate their acquisition of health literacy skills.]
